# A Multi-Level Distributed Computing Approach to XDraw Viewshed Analysis Using Apache Spark

**Junduo Dong** and **Jianbo Zhang** *

School of Geography Information Engineering, China University of Geosciences, Wuhan 430074, China
* Correspondence: zhangjb@cug.edu.cn

**Abstract:** Viewshed analysis is a terrain visibility computation method based on the digital elevation model (DEM). With the rapid growth of remote sensing and data collection technologies, the volume of large-scale raster DEM data has reached a great size (ZB). However, the data storage and GIS analysis based on such large-scale digital data volume become extra difficult. The usually distributed approaches based on Apache Hadoop and Spark can efficiently handle the viewshed analysis computation of large-scale DEM data, but there are still bottleneck and precision problems. In this article, we present a multi-level distributed XDraw (ML-XDraw) algorithm with Apache Spark to handle the viewshed analysis of large DEM data. The ML-XDraw algorithm mainly consists of 3 parts: (1) designing the XDraw algorithm into a multi-level distributed computing process, (2) introducing a multi-level data decomposition strategy to solve the calculating bottleneck problem of the cluster's executor, and (3) proposing a boundary approximate calculation strategy to solve the precision loss problem in calculation near the boundary. Experiments show that the ML-XDraw algorithm adequately addresses the above problems and achieves better speed-up and accuracy as the volume of raster DEM data increases drastically.

**Keywords:** viewshed analysis; XDraw; Spark; distributed computing

## 1. Introduction

The viewshed analysis is a spatial approach to obtaining the visible area of one or more specified viewpoints on the terrain. It is a typical situation of spatial analytics based on digital elevation models (DEMs). The viewshed analysis has been used extensively in military decision-making [1], optimal location selection [2], oceanic blue space [3], optimal path planning [4], security monitoring [5,6], and other fields.

As the technology of remote sensing and sensor improves by leaps and bounds, the digital terrain data's volume of the global earth has reached a great size (ZB) [7]. The application of these large-scale earth observation data to geospatial research has been widely and effectively used in global climate change, earthquake disaster prediction and greenhouse effect analysis [8]. However, the data storage and GIS analysis based on such large-scale digital data volume become extra difficult [7,9]. As a typical GIS analysis case, viewshed analysis with large terrain data also faces huge challenges. On the one hand, traditional GIS software is unable to handle viewshed analysis computation of such large-scale terrain data. On the other hand, the approaches of high-performance computing (HPC) are widely used to accelerate GIS computation-intensive algorithms [10]. It shows great potential to advance parallel spatial analysis [11–13].

Viewshed analysis computes all points' visibility on the terrain [14] with the given observation point. Typically, the line-of-sight (LoS) algorithm judges whether a specific point on terrain is visible by calculating if the straight ray between it and the observation is obscured. According to different model types of terrain, viewshed algorithms are mainly divided into TIN (triangular irregular network)-based algorithms and grid-based

algorithms [15]. The representative TIN-based viewshed algorithms are the hidden surface removal algorithm[16] and Whitted-Style ray tracing-based algorithm [17]. These algorithms assume that the targets are ordered by depth from the viewpoint, and their TIN-based computations have higher accuracy but higher computational complexity. The representative grid-based viewshed algorithms are the R3, R2, and XDraw algorithms [18]. These algorithms' time and space complexities are given in Table 1, where $n$ is the number of points in the DEM. The R2 and XDraw algorithms are slightly less accurate, as they include approximation processing to speed up the total computation.

**Table 1.** The time and space complexities of three grid-based viewshed algorithms.

| Algorithm | Time Complexity | Space Complexity |
|:---:|:---:|:---:|
| R3 | $O(n^3)$ | $O(n)$ |
| R2 | $O(n^2)$ | $O(n^2)$ |
| XDraw | $O(n^2)$ | $O(n)$ |

The R3 algorithm calculates the LoS of every point on the grid from the viewpoint. It is relatively accurate but the steep time cost [19,20]. The R2 algorithm calculates inner points' visibility approximately during the points' LoS computing on boundary [15], so that all points' visibility on the grid is solved after computing all the boundary points' visibility. The XDraw algorithm calculates from the observation point to the outer ones [21], reusing the inner points' LoS computing results. The XDraw algorithm spends less time cost than the R3 and uses fewer memory spaces than the R2 according to Table 1. More significantly, its spatial independence makes it possible to optimize the algorithm to attain a high level of parallelization. Hence, the XDraw algorithm is a better choice for the viewshed analysis.

The viewshed algorithms are usually optimized by multi-core parallel computing with a single machine. For the R3 and R2 algorithms, most research tries to accelerate their LoS calculation by graphics processing units (GPU) [22] and partition-based approach [23]. The I/O efficiency of the R3 and R2 algorithm can be improved by two-layered computing [24], two-level data decomposition [25] and tile-based storage [26] under the GPU's memory limit. The data management can be improved by managing variant memory types (constant, texture, local) in GPU [27]. The data transfer overhead between CPU and GPU can be optimized by multiple command queues [28]. For the XDraw algorithm, its iterative computation-intensive steps can be accelerated by GPU [29] and SIMD [30]. The I/O efficiency of the XDraw algorithm can be optimized by the decomposition of the DEM grid and indexing data with Morton order [30]. The DEM data's management of the XDraw algorithm can be improved in database storage cases [21].

Meanwhile, other research focuses on optimizing viewshed algorithms based on the distributed cluster with multiple machines. The distributed algorithms are usually implemented by multiple machines' collaborative computing, which can process larger terrain data that a single machine cannot process. Compared with the R3 and R2 algorithms, the XDraw algorithm's distributed improvement has been studied more. The improved distributed XDraw algorithm covers 3 parts: the data decomposition strategy, the distributed schedule strategy, and the distributed storage strategy. The data decomposition strategy of the XDraw algorithm splits the data for multiple machines to read and compute. Its usual implementations are equal angle decomposition strategy [18] and its varieties [31,32] with higher accuracy, and equal area decomposition strategy [33] with fast computing speed. The distributed schedule strategy of the XDraw algorithm schedules the total computing process to multiple machines in the distributed cluster. It can be implemented customarily based on OpenMPI [34] such as a fine-granularity scheduling mechanism [35] and a fault-tolerate mechanism [36], or just used the Apache Spark [37]'s own scheduling mechanism [38,39], which is a big data distributed computing framework. The distributed storage strategy can be two-level storage [40], tile-based storage [41] and group-based storage [42] based on Hadoop distributed file system (HDFS) [43] to store the raster DEM data.

The mature distributed computing mechanism of Apache Spark supports the stable implementation and execution of production workload algorithms. Hence, a Spark-based implementation of viewshed analysis makes a lot of sense. The current distributed XDraw computing schemes can handle viewshed computing of large DEM data efficiently by the equal area decomposition strategy and Spark-based computing approach [38]. However, two problems still exist based on these strategies shown in Figure 1:

(1) *The calculating bottleneck problem of the cluster's executor* is that a single executor in the cluster cannot easily hold the decomposed data in limited memory and process. The hardware resources such as CPU and memory in the Spark cluster are divided finely and wrapped into *executors*. The usual distributed approach divides the DEM grid into multiple triangles, and then multiple executors in cluster read and process the data in triangles' MBR (Minimum Bounding Rectangle) in parallel. However, the MBR amplifies the data size of its corresponding triangle, which may not be easily read by a single executor all at once with limited hardware resources. More importantly, the large triangle may not be split further to reduce its MBR's size. As an example shown in Figure 1, the data in both the triangle *a* and its further split part triangle *c* may not be processed all at once by a single executor.

(2) *The precision loss problem in calculation near the boundary* is that the visibility results of grid points near the boundary may not be calculated. In Figure 1, the visibility calculation of point C near the boundary depends on the visibility results of A and B. However, unfortunately, the machine holding point C cannot obtain the calculation result of another machine holding point A, so its visibility cannot be calculated.

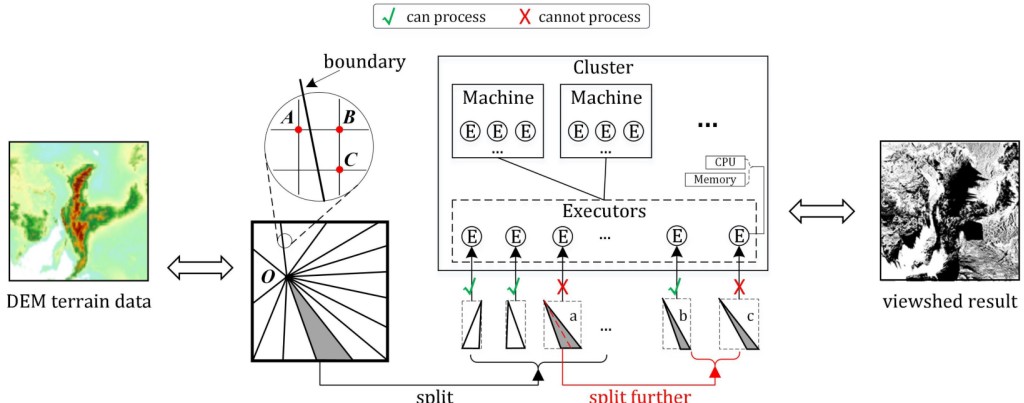

**Figure 1.** The distributed XDraw algorithm with equal area decomposition strategy based on Spark, *O* is the viewpoint.

This article introduces a multi-level distributed XDraw algorithm using Spark to optimize the 2 problems above. The main original contributions are as follows.

(1) An improved approach named multi-level distributed XDraw (ML-XDraw) algorithm is designed to process viewshed analysis of large DEM data with Spark.

(2) A multi-level data decomposition strategy is introduced to solve the calculating bottleneck problem of the cluster's executor.

(3) A boundary approximate calculation strategy is proposed to solve the precision loss problem in calculation near the boundary.

The implementation of the improved XDraw algorithm contains the steps below.

(1) Dividing the DEM grid into multiple levels each holding further divided raster fragments by the multi-level data decomposition strategy.

(2) Calculating each raster fragment's visibility result by the raster fragment-based XDraw algorithm, whose implementation is based on the boundary approximate calculation strategy.

(3)  Organizing the total calculation process into a multi-level distributed algorithm using Apache Spark.

The article is organized into the following sections. In Section 2, the serial XDraw algorithm is introduced, and the principle of the ML-XDraw algorithm, the two strategies above, and the related algorithm's implementation are described. In Section 3, the designed experiments are presented with the relevant evaluation details of performance and accuracy. The experimental results are discussed in detail in Section 4. In Section 5, the key conclusions drawn and additional future research are outlined.

## 2. Methods

### 2.1. Serial XDraw Algorithm

The XDraw is a serial viewshed algorithm that calculates all the DEM grid points' visibility by nested rings' order from inner to outer. All grid points are grouped into multiple rings of width 1 $\{r_1, r_2, \dots\}$ as shown in Figure 2, starting from the viewpoint and working from the inside out.

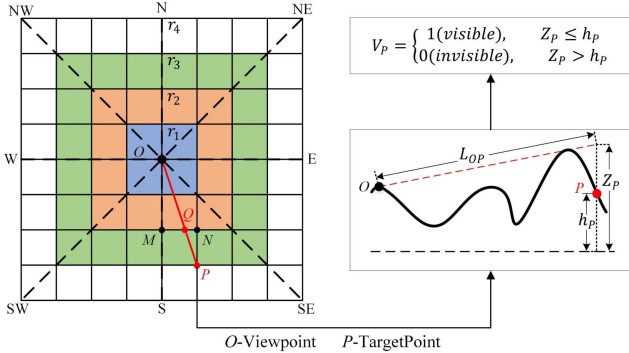

**Figure 2.** Principle of the Serial XDraw algorithm.

For any point $P$ on the terrain, $P$ is visible if and only if $O$ and $P$ can be connected with a straight line without being obscured. $Z_P$ is the minimum height of $P$ which can make it visible, $V_P$ is the visibility of $P$, $h_P$ is the elevation of $P$, $r_P$ is the ring on which $P$ is located, and $L_{OP}$ is the length of line segment $OP$. For any point $P$ on the $r_P$, there must exist 2 grid points $M$ and $N$ on the adjacent inner ring such that $OP$ intersects the line $MN$ at $Q$. $LoS(P)$ can be defined as below:

$$LoS(P) = \begin{cases} h_P, & \text{if } r_P \text{ is } r_1 \\ (interp(Z_M, Z_N) - h_o) \times \frac{L_{OP}}{L_{OQ}} + h_o, & \\ & \text{if } r_P \text{ is not } r_1 \end{cases} \tag{1}$$

where $interp$ can be linear interpolation, so that

$$\begin{cases} Z_P = LoS(P) \\ V_P = \begin{cases} 1, & Z_P \leq h_P \\ 0, & Z_P > h_P \end{cases} \end{cases} \tag{2}$$

$P$ is visible ($V_P = 1$) when $Z_P \leq h_P$, otherwise $P$ is not visible ($V_P = 0$). After that $Z_P$ should be updated by $max(Z_P, h_P)$. The viewshed calculation of the outer ring's grid points depends on the result of the adjacent inner ring ones. The XDraw algorithm solves all the grid points' $V_P$ by iterative calculation from the inner to the outer ring.

## 2.2. Principle of Multi-Level Distributed XDraw Algorithm

### 2.2.1. Overview of the Improved Algorithm

The multi-level distributed XDraw (ML-XDraw) algorithm splits the serial XDraw algorithm into multiple levels' parallel computing process, as shown in Figure 3.

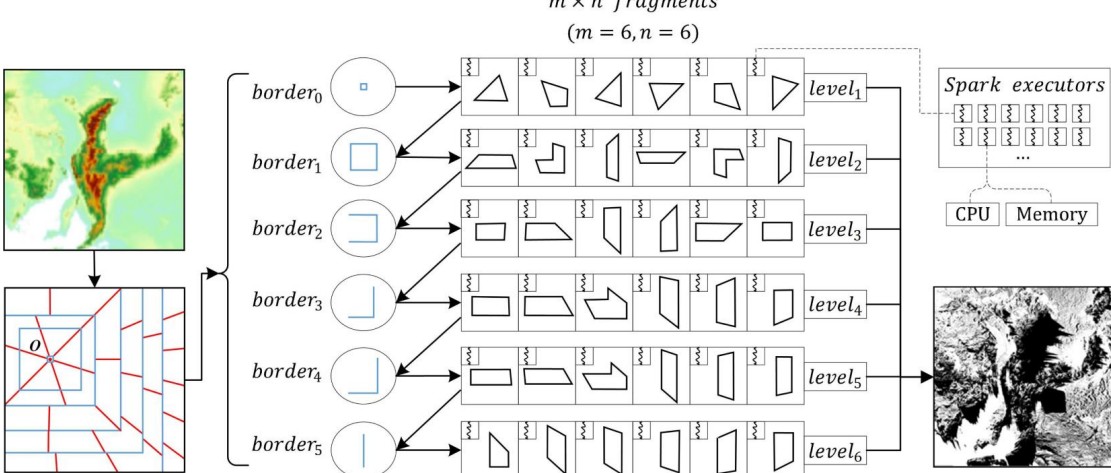

**Figure 3.** Principle of the ML-XDraw algorithm.

First, the multi-level data decomposition strategy divides the DEM grid into $m$ viewpoint-centered annular bands $\{level_1, \ldots level_k, \ldots level_m\}$. The $border_k$ is the ring between $level_k$ and $level_{k+1}$ ($border_0$ is the adjacent ring of viewpoint). The $border_k$ contains up to 4 sub-borders $up, down, left, right$ in the corresponding directions, and the vertical distance between viewpoint $O$ and all sub-borders of $border_k$ is equal. Each level is further divided into $n$ fragments with approximately the same area.

Then, the ML-XDraw algorithm calculates the viewshed results of all levels in turn from $level_1$ to $level_k$ and merges them uniformly. All grid points' $Z_P$ on $border_0$ are their elevation $h_P$. The ML-XDraw algorithm starts from the viewshed computation of $level_1$ driven by $border_0$, and the viewshed computation of $level_{k+1}$ is driven by $border_k$. When $V_P$ of the viewshed result of $level_k$ has been calculated, $Z_P$ of all grid points on $border_k$ has also been calculated. The algorithm combines all intermediate results of every level into the final viewshed result.

Finally, the viewshed computation of the $level_k$ can be accelerated by $n$ fragment-based XDraw algorithm processes in parallel. The $level_k$ is divided into $n$ corresponding fragments, so all grid points' visibility in a single fragment can be individually calculated by a single spark executor, which holds hardware resources of CPU and memory. The $n$ executors do the fragment-based XDraw algorithm concurrently, whose implementation is based on the boundary approximate calculation strategy.

### 2.2.2. Multi-Level Data Decomposition Strategy

The multi-level data decomposition strategy solves the calculating bottleneck problem. This strategy divides the big terrain grid into $m \times n$ raster fragments under the executor memory limit, where $m$ denotes the number of levels, and $n$ denotes the number of divisions with a single level. The flowchart of the data multi-level data decomposition strategy is illustrated by Figure 4. The $S_{max}$ is the maximum raster area that can be analyzed by a single executor. The $border_{end}$ is the DEM gird's out border. The $H_k$ of $level_k$ is the vertical distance between $border_{k-1}$ and $border_k$, and the $fragments_k$ is the set of $n$ fragments created by the division of $level_k$, where $1 \leq k \leq m$.

This strategy calculates the $\{H_k, fragments_k\}$ in numbered order. The function $H$ is used to calculate $H_k$, and the function $F$ is used to divide $level_k$ into $fragments_k$. The strategy firstly calculates $\{H_1, fragments_1\}$ of $level_1$ by $border_0$, and then $border_1$ can be

calculated by $border_0$ and $H_1$. By analogy, $border_k$ can be calculated by $border_{k-1}$ and $H_k$, and then calculate $\{H_{k+1}, fragments_{k+1}\}$ of $level_{k+1}$. The decomposing process ends when $border_k$ attaches $border_{end}$, then the result fragments can be obtained.

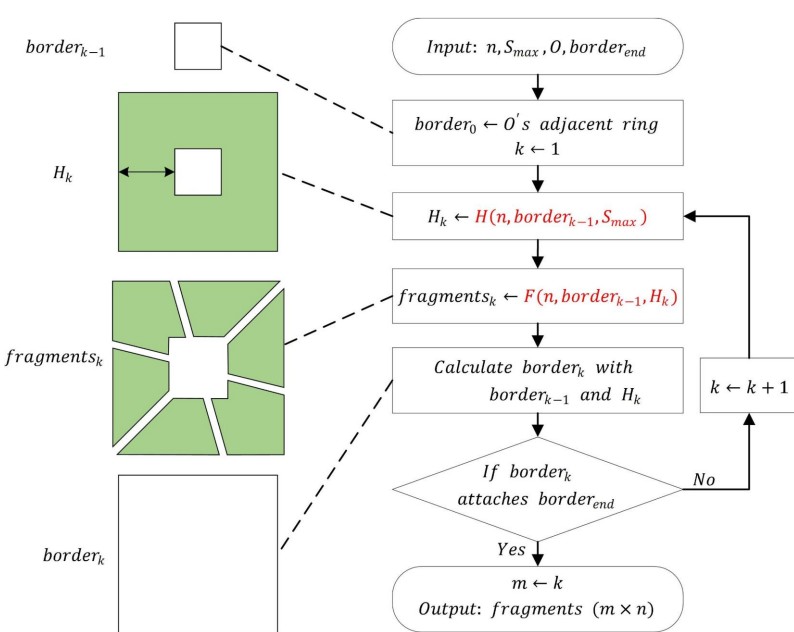

**Figure 4.** The flowchart of the Multi-level Data Decomposition Strategy.

The function $F$ divides the $level_k$ to $n$ area-equal fragments $fragments_k$. The time complexity of the XDraw algorithm is $O(n^2)$ where $n$ is the area (as known as grid points' number) of the grid. The $fragments_k$'s viewshed result can be calculated by $n$ executors in parallel, so all its fragments' area should be equal. Specifically, $border_{k-1}$ is divided counterclockwise $n$ equal parts according to its length, and the lines connecting the $n$-section points and the viewpoint divide $level_k$ into $n$ fragments. Figure 5 shows the details of the $F$ division process, where $m = 6$, $n = 6$. According to the geometric similarity relationship, the areas of all fragments are approximately equal. To facilitate the geometric description of the irregularly shaped fragment, a fragment can be further divided into a set of sectors based on the border's inflection points. The example in Figure 5 shows that a single fragment can be described by a combination of 3 basic sectors, where $a, c \in \{0, 1\}$, $0 \leq b \leq 4$. The $n$-division of $level_k$ is only related to the n, $border_{k-1}$ and $H_k$, so $fragments_k$ is solved by $F$:

$$fragments_k = F(n, border_{k-1}, H_k) \tag{3}$$

The function $H$ calculates the $level_k$'s width $H_k$ with the $S_{max}$ constraint. All sectors in a single fragment are read and computed by a single executor in MBR form. Therefore, the area sum $S$ of all these sectors' MBR should satisfy $S \leq S_{max}$. The length of $border_{k-1}$ is $bl$, the vertical distance between $border_{k-1}$ and viewpoint $O$ is $h'$, the maximum width of $level_k$ is $h$, the sum of the top and bottom edge lengths of all the sectors in a single fragment is $edge$, $edge'$, respectively. Figure 6 defines the other related edges ($le$, $le'$, $re$, $re'$, $e_i$, $e_i'$, $lw$, $lw'$, $rw$, $rw'$) of the 3 basic sectors. The following Equations (4)–(6) exist according to the geometric relationship:

$$bl = \begin{cases} 8, & k = 0 \text{ and } O = O_1 \\ 4, & k = 0 \text{ and } O = O_2 \\ 2, & k = 0 \text{ and } O = O_3 \\ border_k\text{'s length}, & k > 0 \end{cases} \tag{4}$$

$$\begin{cases} edge = le + \sum_{i=1}^{b} e_i + re \\[2mm] edge' = le' + \sum_{i=1}^{b} e_i' + re' = \dfrac{bl}{n} \\[2mm] \dfrac{le}{le'} = \dfrac{e_i}{e_i'} = \dfrac{re}{re'} = \dfrac{h'+h}{h'} \\[2mm] \dfrac{lw}{lw'} = \dfrac{rw}{rw'} = \dfrac{h}{h'} \\[2mm] S = (edge + lw + rw) \times h \end{cases} \tag{5}$$

$$S_{max} = max(S) \tag{6}$$

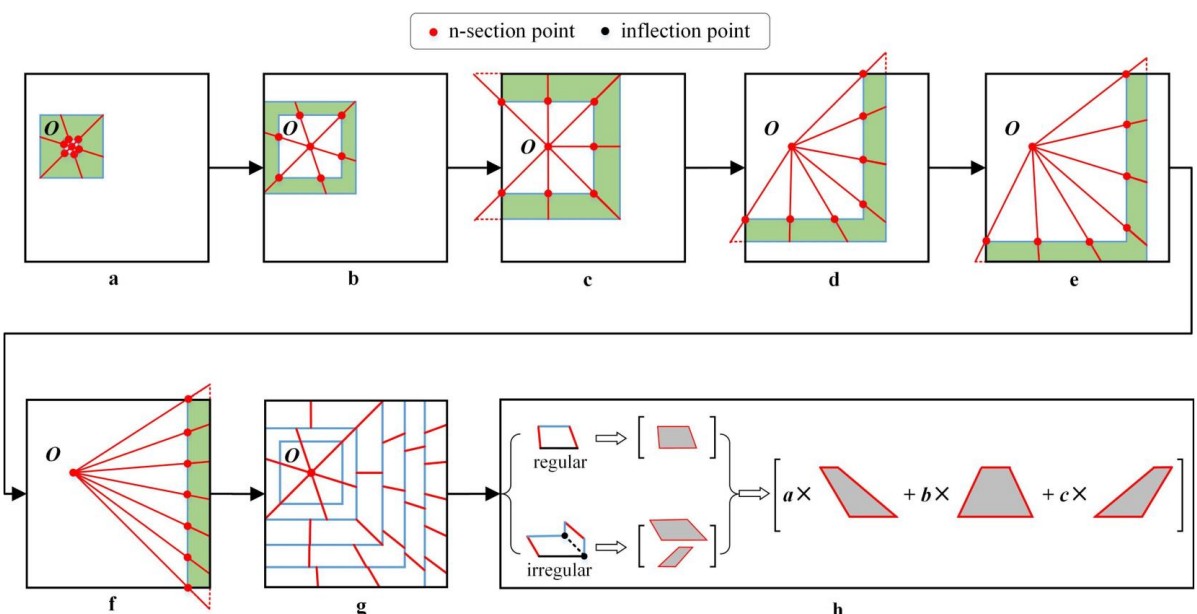

**Figure 5.** The details of function *F* in the Multi-level Data Decomposition Strategy ($m = 6$, $n = 6$, $a, c \in \{0, 1\}$, $0 \le b \le 4$). (**a**–**g**) are the divisions of ($level_1$, $level_2$, $level_3$, $level_4$, $level_5$, $level_6$), and (**h**) is the details of the decomposed fragments.

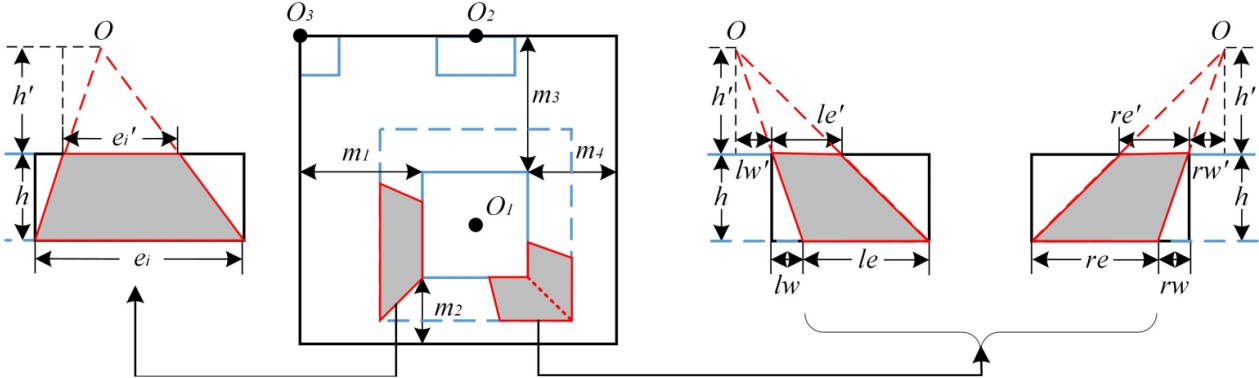

**Figure 6.** The details of function *H* in the Multi-level Data Decomposition Strategy.

The $h$ can be solved and defined as $f$ based on the equations above.

$$\begin{cases} h = \dfrac{-\dfrac{bl}{n} + \sqrt{(\dfrac{bl}{n})^2 + 4 \times (\dfrac{\dfrac{bl}{n} + \Delta}{h'} \times S_{max})}}{2 \times \dfrac{\dfrac{bl}{n} + \Delta}{h'}} \\ = f(n, bl, \Delta, h', S_{max}) \\ \Delta = max(lw' + rw') \end{cases} \quad (7)$$

$\Delta$ is the maximum value of all the fragments' $lw' + rw'$ in $level_k$. The $\Delta$ can be easily calculated by traversing the geometric positions of $n$-section points and inflection points of $border_{k-1}$. The $m_1, m_2, m_3, m_4$ are defined as the margin from $border_{k-1}$'s sub-border $left$, $down$, $up$ and $right$ to the $border_{end}$, respectively. The $H_k$ should be the minimum of $h, m_1, m_2, m_3, m_4$, so that:

$$\begin{aligned} H_k &= min(h, m_1, m_2, m_3, m_4) \\ &= min(f(n, bl, \Delta, h', S_{max}), m_1, m_2, m_3, m_4) \\ &= H(n, border_{k-1}, S_{max}) \end{aligned} \quad (8)$$

### 2.2.3. Boundary Approximate Calculation Strategy

The boundary approximate calculation strategy solves the precision loss problem. The example in Figure 7 shows that the LoS calculation of point F on sector boundary relies on $Z_D$ and $Z_E$, but D is divided into another sector resulting in the inability to solve for $Z_F$. For this reason, this strategy implements an approximation approach by calculating one more grid point's $Z_P$ on the left or right boundary. Taking the left boundary in Figure 7 as an example, $Z_M$ of the intersection point $M$ which is created by the left boundary and BC can be calculated by $Z_A$, then update $Z_B$ with $Z_M$. For the next row's iteration, $Z_N$ of the intersection point $N$ can be calculated by $Z_B$ and $Z_C$, then update $Z_D$ with $Z_N$. Finally, the $Z_F$ can be calculated by $Z_D$ and $Z_E$.

The raster fragment-based XDraw algorithm is implemented based on the boundary approximate calculation strategy. An executor calculates a raster fragment's visibility result by this algorithm. The main process is traversing a fragment's sectors and processing them with the XDraw algorithm, which is given in Figure 7. The fragment in Figure 7 contains 2 sectors a and b. The algorithm performs iterative computation from top to bottom for a and b, respectively, to obtain the $Z_p$ and $V_P$ of the grid points. First, the algorithm copies $border_k$'s corresponding grid points' $Z_p$ into the array buckets. Then, the algorithm calculates the grid points' $Z_p$ and $V_P$ row by row. $V_P$ will be saved to the viewshed bitmap, and $Z_p$ will update the corresponding grid points' $Z_p$ in array buckets. Finally, the algorithm copies the grid points' $Z_p$ stored in buckets and updates the corresponding grid points' $Z_p$ of $border_{k+1}$.

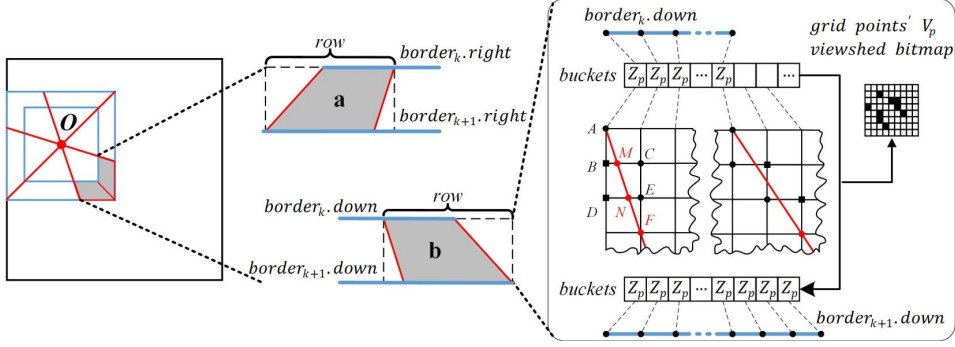

**Figure 7.** The Boundary Approximate Calculation Strategy.

*2.3. Algorithms Implementation Based on Spark*

The Spark-based ML-XDraw algorithm is implemented based on the multi-level data decomposition strategy, which is the overall distributed computing process. The raster fragment-based XDraw algorithm is implemented based on the boundary approximate calculation strategy, which is executed by a single executor to calculate a single raster fragment's visibility result.

2.3.1. Spark-Based ML-XDraw Algorithm

The ML-XDraw algorithm is a level-grained iterative computation, and it can be implemented by Apache Spark to accelerate its parallel process. Figure 8 gives the details of the overall computation. The total process includes 3 main stages: the build stage, the submit stage, and the merge stage. The build stage builds all levels into RDDs (Resilient Distributed Datasets). Each RDD treats a level's fragments as multiple partitions which can be calculated in parallel. The submit stage submits RDDs in the group to the Spark cluster to execute actually. A partition's computation should be allocated to an executor. If each RDD holds $n$ partitions, no more than w RDDs can be submitted when there are $e(\approx w \times n)$ executors. The merge stage merges all executors' cached computed results to the final visibility bitmap.

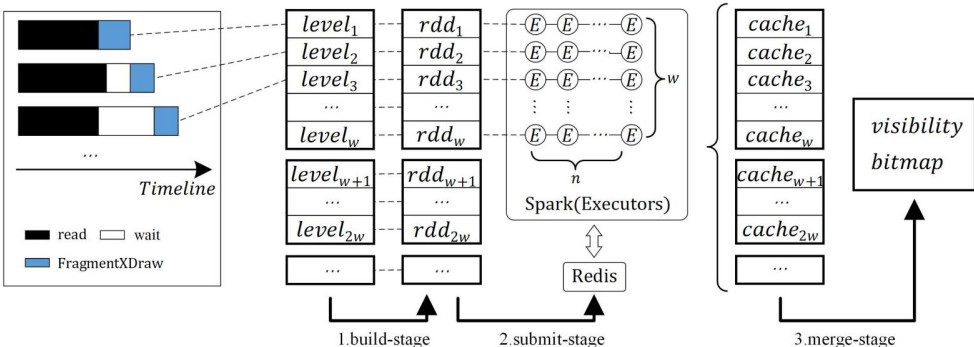

**Figure 8.** The detail of Spark-based ML-XDraw, which submits multiple RDD tasks concurrently.

The details of the ML-XDraw algorithm are described in Algorithm 1. The operator *map(func)* returns a newly formed dataset by passing elements in the origin dataset through the function *func*. The operator *reduce(func)* aggregates the elements of the dataset using the function *func* (which accepts two elements and outputs one). The operator *cache()* persists in a dataset and keeps it in the memory of executors. The inputs of the ML-XDraw algorithm are raster DEM data $D$, viewpoint $O$, grid's border $border_{end}$, and the cluster's resource information. Step 1 uses the multi-level data decomposition strategy to decompose $D$ into $m \times n$ fragments. The build stage contains steps 3–9. Step 5 adds a read operator to read fragments[k][j] into $rdd_k$'s partition j created by step 4. Step 6 blocks the process until $rdd_k$ gets the $border_{k-1}$. Steps 7–8 add the FragmentXDraw operator to calculate the visibility result of fragments[k][j] and cache it. Steps 10–11 calculate the maximum amount of submitted RDDs $w$, get and store $border_0$. The submit stage contains steps 12–23. Steps 16–18 use map-reduce operators to merge all partitions' results of $rdd_k$ into $border_k$ and store it in Redis. After that, the $rdd_{k+1}$'s calculation can be unblocked in Step 6. Step 22 submit RDDs in groups of $k$. The merge stage with steps 24–27 merges all RDD's visibility results to the final result and writes it for storage.

---

**Algorithm 1:** Spark-based ML-XDraw algorithm

---

> **input** : the raster DEM $D$, the viewpoint $O$, the end border $border_{end}$, the Redis
> instance $R$, the SparkContext instance $sc$ with $e$ available executors, the
> number of divisions $n$, the max raster area each executor can handle $S_{max}$
>
> **output**: viewshed result bitmap

**1** Split $D$ to an array $fragments$ of size $m \times n$ by multi-level data decomposition
  strategy with $\{n, S_{max}, O, border_{end}\}$

**2** $rdds \leftarrow$ `Array[RDD[border,bitmap]]`$(m)$;

**3 for** $k \leftarrow 1$ **to** $m$ **do**                                                       `// 1.build stage`

**4**  $\quad$ $rdds[k] \leftarrow sc$ `.parallelize(`$[1, 2, ..., n]$`)`

**5**  $\quad$ `.map(`$j \rightarrow$ `read(`$fragments[k][j]$`))`

**6**  $\quad$ `.map(`**repeat** $b \leftarrow R$`.get(`$k-1$`)` **until** $b \neq null$`)`

**7**  $\quad$ `.map(`$fgmt \rightarrow$ `FragmentXDraw(`$fgmt, b$`))`

**8**  $\quad$ `.cache();`

**9 end**

**10** $w \leftarrow e \,/\, n$, $border_0 \leftarrow O$ 's adjacent ring;

**11** $R$`.set(`$0, border_0$`)`;

**12 for** $start \leftarrow 1$ **to** $m$ **with step** $w$ **do**                                `// 2.submit stage`

**13**  $\quad$ $tasks \leftarrow$ `Empty Set`;

**14**  $\quad$ **for** $k \leftarrow start$ **to** $start + w$ **do**

**15**  $\quad\quad$ $tasks$`.add(`$\{$

**16**  $\quad\quad$ $nborder \leftarrow rdds[k]$

**17**  $\quad\quad\quad$ `.map((border,bitmap)` $\rightarrow$ `border)`

**18**  $\quad\quad\quad$ `.reduce((`$b1, b2$`)` $\rightarrow$ $b1 + b2$`)`;

**19**  $\quad\quad$ $R$`.set(`$k, nborder$`)`;

**20**  $\quad\quad$ $\}$`)`;

**21**  $\quad$ **end**

**22**  $\quad$ submit $tasks$ concurrently to Spark

**23 end**

**24** $sc$`.union(`$rdds$`)`                                                                    `// 3.merge stage`

**25**  $\quad$ `.map((border,bitmap)` $\rightarrow$ `bitmap)`

**26**  $\quad$ `.reduce((`$bm1, bm2$`)` $\rightarrow$ `merge(`$bm1, bm2$`))`

**27**  $\quad$ `.write();`

---

### 2.3.2. Raster Fragment-Based XDraw Algorithm

The Raster Fragment-based XDraw algorithm consists of two main parts: *FragmentXDraw* and *SectorXDraw*. The FragmentXDraw is the main entry algorithm. The input of this algorithm is the raster fragment $fgmt$ and its up-border $border$, and the output is the viewshed bitmap $nbitmap$ and down-border $nborder$ of $fgmt$. The up-border and down-border of a fragment in $level_k$ are $border_{k-1}$, $border_k$, respectively.

The details of the FragmentXDraw are described in Algorithm 2. Steps 1–2 initialize the $nbitmap$, $nborder$, and iteration array buckets returned by the algorithm. Steps 3–9 traverse $fgmt$'s all the sectors and calculate with the XDraw algorithm. First, step 5 is responsible for copying the corresponding grid points' $Z_p$ in the up-order to the buckets. Second, step 6 is responsible for executing the SectorXDraw algorithm to obtain the viewshed result for the sector. Then, step 7 is responsible for copying the buckets' $Z_p$ to the corresponding grid points on the down-border. Finally, step 8 is responsible for merging the viewshed result of the sector with the final result of the fragment it belongs to.

The details of the SectorXDraw are described in Algorithm 3. It traverses row by row from the viewpoint and calculates the visibility of each grid point inside the sector in turn. More details of the algorithm are described in Algorithm 3. Step 1 initializes the temporary

array zp and the viewshed bitmap bm. Steps 4–5 get the left and right boundaries of the sector, ignoring the grid points beyond the sector boundary. Steps 6–19 calculate the elevation of the left boundary point, right boundary point, and interior point of this sector, respectively, and calculate the $Z_p$ and $V_p$ of each point by LoS algorithm. The zp keeps the $Z_p$ of the grid points in this row, and the bm keeps the $V_p$ of all grid points. Step 20 copies that row's zp to the array buckets for the next XDraw iteration.

---

**Algorithm 2:** FragmentXDraw

**input** : the raster fragment data *fgmt*, the up-border *border*
**output**: the down-border *nborder*, all grid points' $V_p$ bitmap in that fragment *nbitmap*

1 *nborder*, *nbitmap* ← empty border, bitmap;
2 *buckets* ← empty float array;
  // sectors[i].d="left|right|up|down"
3 **foreach** *sector* ← *fgmt.sectors* **do**
4      $d$ ← *sector.d*;
5      copy the grid points' $Z_p$ in *border*[$d$] to *buckets*;
6      *buckets*, *bm* ← SectorXDraw(*sector*, *buckets*);
7      copy *buckets* to the grid points' $Z_p$ in *border*[$d$];
8      *nbitmap* ← merge(*nbitmap*, *bm*);
9 **end**
10 **return** *nborder*, *nbitmap*;

---

**Algorithm 3:** SectorXDraw

**input** : the sector's MBR DEM data *sector*, all the first row grid points' $Z_p$ array *buckets*
**output**: all the last row grid points' $Z_p$ array *buckets*, all grid points' $V_p$ bitmap in that sector *bm*

1 *zp* ← empty float array, *bm* ← empty bitmap;
2 **for** $i$ ← 1 **to** *sector.height* **do**
3      **for** $j$ ← 1 **to** *sector.width* **do**
4          le, rt ← *sector*[$i$].*leftJ*, *sector*[$i$].*rightJ*;
5          **if** $j <$ le **and** $i >$ rt **then continue**;
6          **if** $j =$ le.*floor* **then**        // left boundary
7             $hp$ ← (le − $j$) ∗ *sector*[$i$][$j$]+
8                 ($j + 1 −$ le) ∗ *sector*[$i$][$j + 1$];
9             $zp$[$j$] ← LoS($i$, le, *buckets*);
10          **else if** $j =$ rt.*ceil* **then**      // right boundary
11             $hp$ ← ($j + 1 −$ rt) ∗ *sector*[$i$][$j + 1$]+
12                 (rt − $j$) ∗ *sector*[$i$][$j$];
13             $zp$[$j$] ← LoS($i$, rt, *buckets*);
14          **else**                         // inner
15             $hp$, $zp$[$j$] ← *sector*[$i$][$j$], LoS($i$, $j$, *buckets*)
16          **endif**
17          $bm$[$i$][$j$] ← **if** $zp$[$j$] $\le hp$ **then** 0 **else** 1;
18          $zp$[$j$] ← max($zp$[$j$], $hp$);
19      **end**
20      **for** $j$ ← 1 **to** *sector.width* **do** *buckets*[$j$] ← $zp$[$j$];
21 **end**
22 **return** *buckets*, *bm*

## 3. Experiments and Results

### 3.1. Datasets

Open access DEM data of Australia (Available: https://data.gov.au/data/dataset/9a9 284b6-eb45-4a13-97d0-91bf25f1187b, (accessed on 1 February 2020)) was used to evaluate the computational performance, accuracy, and distributed scalability of the ML-XDraw algorithm. Figure 9 is the elevation map and the histogram of Australia. The elevation of most areas is located in the interval [0, 600] (unit: meter).

The determining factors that affect the computation of the viewshed algorithm are the volume of DEM data and the location of the viewpoint. To verify the efficient processing capability of ML-XDraw on large-scale DEM data, the experiments were designed based on four elevation datasets with different sizes created by the origin DEM's resample. The associated data files are stored in the extended GRD format and are described in Table 2. The grid cells located in Australia are the valid points in the dataset DEM grid, and the others are invalid. To determine that ML-XDraw achieves good performance under various viewpoint locations, three different types of viewpoints (points on the pit, peak, and flat areas) were chosen in the corresponding experiments.

**Table 2.** Experimental datasets.

| DataSet | Size (GB) | Grid-Cell Size | Columns | Rows | Total Grid Cells | Valid Grid Cells |
|---------|-----------|----------------|---------|------|------------------|------------------|
| DEM1 | 16.83 | $2'' \times 2''$ | 73,800 | 61,201 | 4,516,633,800 | 2,634,119,863 |
| DEM2 | 29.91 | $1.5'' \times 1.5''$ | 98,399 | 81,602 | 8,029,555,198 | 4,682,469,269 |
| DEM3 | 67.30 | $1'' \times 1''$ | 147,600 | 122,401 | 18,066,387,600 | 10,536,479,532 |
| DEM4 | 105.16 | $0.8'' \times 0.8''$ | 184,500 | 153,001 | 28,228,684,500 | 16,462,348,871 |

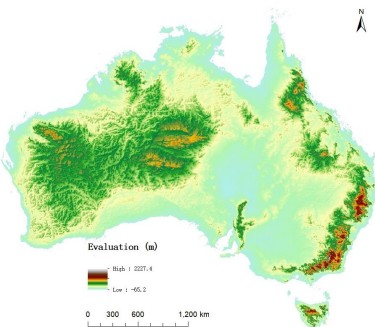

(**a**) Elevation map of Australia.

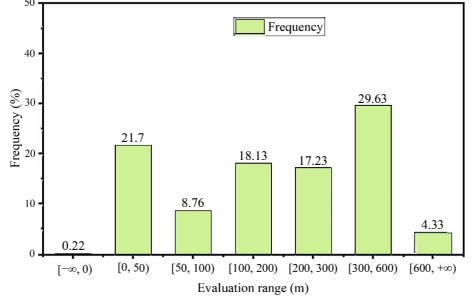

(**b**) Histogram of elevation in Australia.

**Figure 9.** Elevation map and histogram of Australia.

### 3.2. Hardware Environment

An Apache Spark cluster holding seven computers was used as the distributed computing platform for the ML-XDraw algorithm. All the cluster nodes were linked by 1 Gbps rapid ethernet. There were, in all, 96 cores of CPU capacity, 384 GB of memory, and 16 TB HDD of storage in this cluster. A single computer workstation was utilized to perform the serial XDraw algorithm for comparison. It had 48 cores of CPU capacity, 96 GB of memory, and 10 TB HDD of storage. The program compilation environment is JDK-1.8 and Scala-2.11.11. The runtime environment for the algorithm is Ubuntu 18.04, Hadoop 2.7.1, and Spark 2.4.7 on all computers.

### 3.3. Experimental Designs

To test the parallel performance of the ML-XDraw algorithm in this article, four experiments were conducted. Each experiment was repeated five times, and the results were averaged.

(1) To study the effectiveness of ML-XDraw in terms of parallel performance, both the serial XDraw algorithm implemented on a workstation and the ML-XDraw algorithm

implemented on a cluster was tested on four datasets and three viewpoints. Limited by the capability of the serial algorithm to handle big terrain data, both algorithms were executed based on the first three datasets and three types of viewpoints.

(2) To investigate the impact of the multi-level data decomposition strategy on the overall performance of ML-XDraw, the different number of divisions are used to explore how the decomposed results affect the overall performance.

(3) To verify the accuracy improvement of the boundary approximate calculation strategy, the serial XDraw algorithm implemented on the workstation was chosen as a correct reference. Only the first three datasets were chosen for the same reason as (1).

(4) To explore the scale-out distributed performance of the ML-XDraw algorithm, it was performed on four datasets and three viewpoints with an increasing number of Spark executors.

### 3.4. Performance Evaluation

(1) *The speedup ratio (SR)* was used to measure the performance of the ML-XDraw algorithm. It is defined as the ratio between the computational time of the viewshed analysis executed on the workstation and that implemented on the cluster. Its equation is as follows:

$$SR = \frac{T_{serial-XDraw}}{T_{ML-XDraw}} \qquad (9)$$

where $SR$ is the speedup ratio, $T_{serial-XDraw}$ is the computational time of the viewshed algorithm on the workstation, and the $T_{ML-XDraw}$ is the computational time of the proposed algorithm on the cluster.

(2) *The average area ratio (AAR)* was used to measure the efficiency of the multi-level data decomposition strategy. It describes the average area of DEM grid data processed by each executor throughout a distributed computation. The different number of divisions will produce different decomposition results, thus the read amplification introduced by reading the data in MBR type is variant. The AAR is defined as follows:

$$AAR = \frac{\sum A_{MBR}}{A_{DEM} \times N_{executors}} \qquad (10)$$

where $\sum A_{MBR}$ is the area sum of all sectors' MBR decomposed of the multi-level data decomposition strategy, $A_{DEM}$ is the area of the total DEM grid, and $N_{executors}$ is the number of executors in use.

(3) *The correctness ratio (CR)* was used to measure the accuracy of the proposed approach when using the boundary approximate calculation strategy or not. It is defined as follows:

$$CR = \frac{N_{correct}}{N_{total}} \qquad (11)$$

where $N_{total}$ is the total number of valid grid cells in the raster DEM, and $N_{correct}$ is the number of grid cells with the same visibility result between those of the proposed algorithm and those of the serial XDraw algorithm.

### 3.5. Results

The viewshed result maps are illustrated in Figure 10. The two algorithms are tested on the same dataset DEM3 and three viewpoints. Three viewpoints were used to illustrate comprehensively the viewshed results of the two algorithms. The DEM3 was used as the viewshed analysis data to compare the two algorithms. The larger dataset DEM4 cannot be chosen due to the serial XDraw algorithm cannot calculate it on the workstation's limited hardware now. The experimental results and further detailed discussions are given in Section 4.

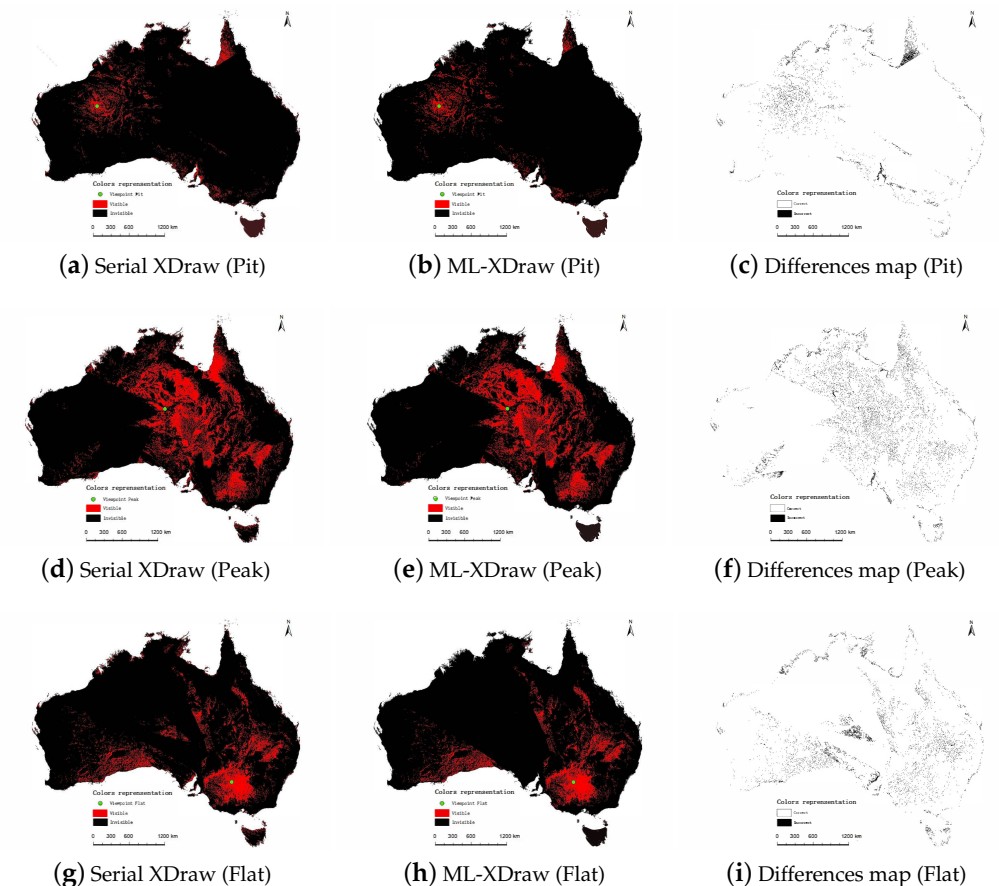

**Figure 10.** Viewshed result maps generated by two algorithms based on dataset DEM3. (**a**,**d**,**g**) are the calculation results of serial XDraw algorithm. (**b**,**e**,**h**) are the calculation results of ML-XDraw algorithm. (**c**,**f**,**i**) are the differences maps of the two algorithms' results.

## 4. Discussions

In Section 4.2, 8 was proved to be the optimal number of divisions for the ML-XDraw algorithm with Spark. Therefore, in Sections 4.1, 4.3, and 4.4, 8 was selected as the number of divisions. Except for the experiment shown in the scale-out performance of the approach, the other three experiments used 64 executors, each with 1 CPU core and 6G RAM. We specified 25% of memory for Spark runtime usage and 75% of memory for data storage. Thus, the maximum area of DEM grid data that a single executor could process was approximately $S_{max} = 1.2 \times 10^9$.

### 4.1. Effectiveness of the Approach on Parallel Performance

The performance results illustrated in Figure 11 and Table 3 show that the ML-XDraw algorithm is computationally efficient as the data volume is increased. First, the multi-level data decomposition strategy has a compounded influence on the acceleration of the XDraw algorithm because it divides the raster DEM into multiple fragments with approximately the same area by maximizing the memory utilization of Spark executors on the cluster. This approach takes full account of the maximum data area that can be handled by a single actuator and is more suitable for computing characteristics of the distributed framework Spark. Second, the ML-XDraw algorithm is relatively gentle under the influence of data volume. As the volume of data increases, the overall computational time of the ML-XDraw algorithm performs steadily, while that of the serial XDraw algorithm increases rapidly. This is because a load of data input is balanced to parallel executors in ML-XDraw, but the total load is concentrated on one computer in the serial algorithm. Multiple nodes equally share the computational burden of increasing the data size. Finally, the variant

viewpoint on the terrain has nearly no effect on the ML-XDraw algorithm but affects the performance of the serial XDraw algorithm. This is because the serial algorithm spreads out the computation from the viewport according to the visibility of the neighboring points. The computational overhead in the different directions is variant in the serial algorithm. Instead, the ML-XDraw algorithm divides all levels around the viewport as evenly as possible, subdivides each level further into the same number of fragments, and calculates them orderly. The computational overhead under three kinds of viewpoints is nearly the same as a result.

**Table 3.** The two algorithms' overall computing time and speedup ratios.

| DataSet | Viewpoint | Serial XDraw (min) | ML-XDraw (min) | Speedup Ratio |
|---------|-----------|--------------------|-----------------|---------------|
| DEM1 | Pit | 17.70 | 4.96 | 3.57 |
| | Peak | 15.66 | 5.84 | 2.68 |
| | Flat | 12.44 | 5.44 | 2.29 |
| DEM2 | Pit | 37.14 | 7.24 | 5.13 |
| | Peak | 42.54 | 8.04 | 5.29 |
| | Flat | 30.64 | 7.46 | 4.11 |
| DEM3 | Pit | 195.80 | 14.89 | 13.15 |
| | Peak | 148.09 | 16.51 | 8.97 |
| | Flat | 137.81 | 16.31 | 8.45 |
| DEM4 | Pit | - | 27.82 | - |
| | Peak | - | 28.81 | - |
| | Flat | - | 32.18 | - |

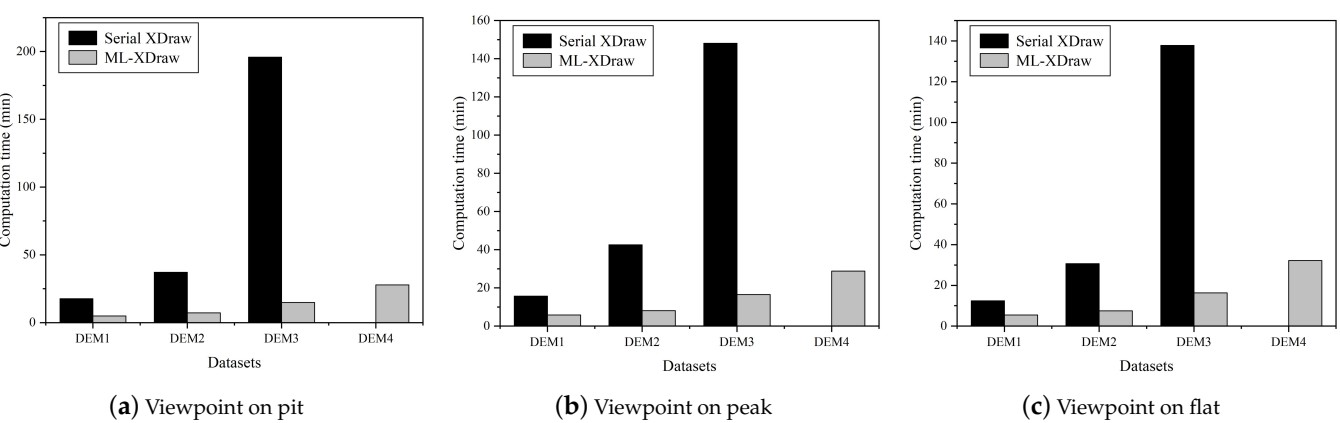

(**a**) Viewpoint on pit  (**b**) Viewpoint on peak  (**c**) Viewpoint on flat

**Figure 11.** Comparisons of the computation time of two algorithms.

*4.2. Effectiveness of Data Decomposition Strategy*

(1) *Eliminating the calculating bottleneck of a single executor:* The existing distributed XDraw algorithms which adopt the equal area decomposition strategy, focus on obtaining evenly-divided areas for the viewshed calculation. With the increase of the data's amount, the calculation area generated by these data decomposition strategies could easily exceed the maximum handling capacity of a single executor. The max area statistics processed by a single executor are shown in Figure 12. It indicates that as the DEM data volume is increased, the data decomposition strategy can steadily maintain the maximum area within the processable range by a single executor (The red line in Figure 11 stands for the maximum area $S_{max}$ that a single executor can process). The reason is that the function $H$ in the multi-level data decomposition strategy calculates each triangle's maximum height

under the max capability ($S_{max}$), so that the area of each triangle's MBR which is processed by its corresponding executor cannot beyond $S_{max}$.

On the contrary, the decomposed part's area without using that strategy may be beyond the $S_{max}$, so that a single executor cannot easily process it. That is because the area of each decomposed triangle's MBR increases without constraints as the dataset volume increases.

(2) *Reducing read amplification by increasing the number of divisions:* The experiment results shown in Figures 13 and 14 indicate that as the number of divisions increases, the calculation time of ML-XDraw becomes shorter and the AAR becomes smaller. Based on the result, 8 was the optimal number of divisions for the ML-XDraw algorithm with Spark. The reason is that the smaller AAR means less read amplification effect caused by the MBR data process approach, and each executor may process fewer data and cost less time. With the increase in the number of divisions, each triangle is finely decomposed of the function $F$. Fine decomposition makes the difference between the triangle and its MBR smaller so that the read amplification introduced by MBR is reduced.

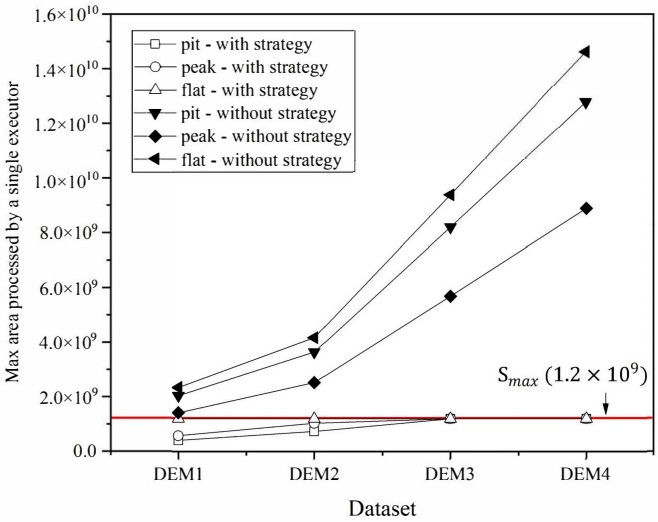

**Figure 12.** Comparisons of max area statistics processed by a single executor with or without the multi-level data decomposition strategy.

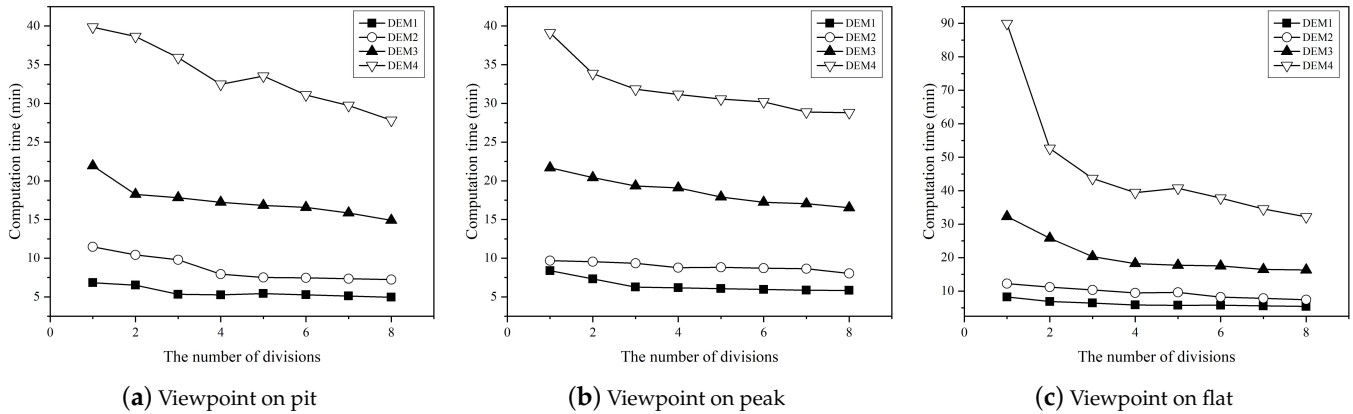

(**a**) Viewpoint on pit　　　　　　(**b**) Viewpoint on peak　　　　　　(**c**) Viewpoint on flat

**Figure 13.** Comparisons of the computation time of different number of divisions.

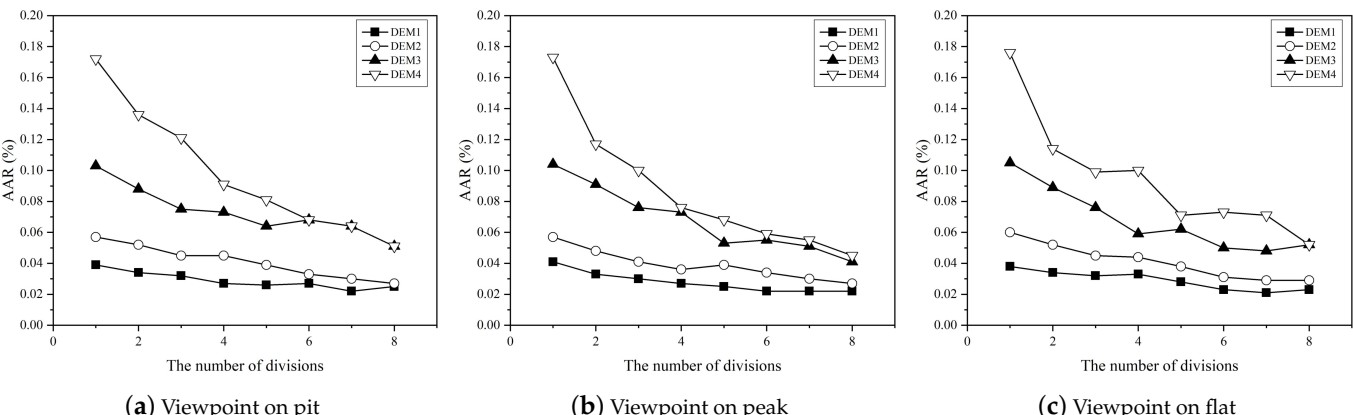

**Figure 14.** Comparisons of the average area ratio of different number of divisions.

### 4.3. Accuracy Comparison of Algorithms

In this experiment, the results' accuracy of ML-XDraw is evaluated in cases with the number of divisions 8 because of its fewer time costs. As an improved approach to XDraw, this experiment uses (serial) XDraw's result as the correctness baseline. DEM1-DEM3 were chosen except DEM4 because the serial XDraw may fail on the workstation with limited resources. Meanwhile, the ML-XDraw without the boundary approximate calculation strategy is compared to explore its effectiveness. The $CR_{with}$ and $CR_{without}$ are ML-XDraw with and without the boundary approximate calculation strategy, respectively.

The results in Table 4 indicate that data size and the number of divisions have no significant effect on the accuracy. However, there are significant results differences between different viewpoint positions. As can be seen from the overall results in Figure 10, the difference between the results of the ML-XDraw and serial XDraw algorithms is more pronounced at the pit viewpoint and peak viewpoint than at the flat viewpoint. The differences in the calculations for the pit viewpoint are mainly in the northeast, while the differences in the calculations for the peak and flat viewpoints are mainly in the east and central regions. This is due to the fact that the terrain around pits and peaks is more complex than flat, so the boundary approximation computation strategy cannot accommodate its drastic terrain changes and ineffectively address the accuracy loss. Therefore, the calculation accuracy of the flat viewpoint in Table 4 is higher than that of the pit viewpoint and peak viewpoint. More importantly, the ML-XDraw with approximation is better than without one, indicating that the approximation strategy works well. The reason is that the terrain of adjacent grid points is similar, so the line of sight (LoS) of grid points can be approximated by using the elevation information of adjacent grid points. The boundary approximate calculation strategy uses the nearby grid points' Zp to calculate the points' visibility result on the boundary approximately (given detailed in Section 2.2.3), instead of just filling it with invisibility result directly.

**Table 4.** Accuracy comparison of the ML-XDraw with the boundary approximate calculation strategy and without one.

| Dataset | Viewpoint | The Number of Correct Grid Cells with or without the Boundary Approximate Calculation Strategy | | The Number of Total Valid Grid Cells | The Correctness Ratio (CR) with or without the Boundary Approximate Calculation Strategy | |
|---|---|---|---|---|---|---|
| | | **With** | **Without** | | $CR_{with}$ | $CR_{without}$ |
| DEM1 | Pit | 2,542,432,197 | 2,521,655,681 | 2,634,119,863 | 96.52% | 95.73% |
| | Peak | 2,557,788,752 | 2,522,559,008 | | 97.10% | 95.76% |
| | Flat | 2,580,371,920 | 2,537,012,236 | | 97.96% | 96.31% |
| DEM2 | Pit | 4,519,469,298 | 4,482,533,344 | 4,682,469,269 | 96.52% | 95.73% |
| | Peak | 4,544,360,919 | 4,484,942,211 | | 97.05% | 95.78% |
| | Flat | 4,584,508,695 | 4,510,636,787 | | 97.91% | 96.33% |
| DEM3 | Pit | 10,169,731,863 | 10,090,239,758 | 10,536,479,532 | 96.52% | 95.76% |
| | Peak | 10,227,544,304 | 10,092,046,397 | | 97.07% | 95.78% |
| | Flat | 10,321,489,519 | 10,155,278,753 | | 97.96% | 96.38% |

*4.4. Scale-Out Performance of the Approach*

In this experiment, the ML-XDraw is carried out on a Spark cluster with different hardware resources. The executor is the computing unit to do the real calculation, so the number of executors determines the total calculation time of the distributed algorithm. Figure 15 shows the experimental result in cases of variant viewpoints, datasets, and the number of executors. First, the computation time of the ML-XDraw decreases as the number of executors increases totally. This is because the more executors there are, the more calculation jobs of ML-XDraw can be submitted. Second, the smaller the DEM dataset, the earlier the curve flattens out. It is due to that the smaller DEM may be divided into fewer raster fragments, so it needs fewer executors to submit and compute all levels completely at once.

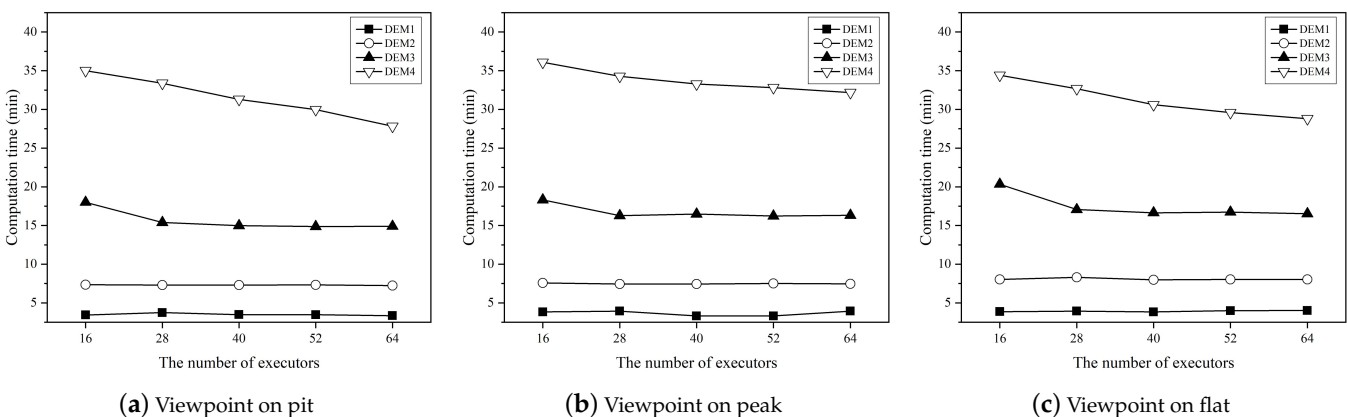

(**a**) Viewpoint on pit      (**b**) Viewpoint on peak      (**c**) Viewpoint on flat

**Figure 15.** Comparisons of the computation time of different number of executors.

## 5. Conclusions

Compared with other viewshed algorithms, the XDraw algorithm is faster and more accurate, so it is more commonly used in single viewpoint viewshed analysis. With the continuous development of more refined remote sensing technology and the expansion of DEM data scale, viewshed analysis based on large-scale DEM becomes particularly important. This article proposes a multi-level distributed XDraw viewshed analysis method

based on Spark, which includes the multi-level distributed XDraw algorithm (ML-XDraw), the multi-level data decomposition strategy, and the boundary approximate calculation strategy. First, the ML-XDraw divides the whole process into multi-level coarse-grained iterative computations, which makes XDraw computation on a large-scale grid suitable for computing clusters with different computing levels. Second, the multi-level data decomposition strategy gives a data division method under the limitation of single executor memory size, which solves the bottleneck problem. Then, the boundary approximate calculation strategy gives the approximate processing method of data boundary, which reduces the precision loss to a certain extent. Finally, this article implements the ML-XDraw based on Spark. The experimental results show that the ML-XDraw has a perfect acceleration effect on the viewshed analysis of large-scale DEM, and has good scalability with small precision loss.

In future work, we will continue to focus on the improvement and optimization of ML-XDraw in the viewshed analysis of large-scale DEM grid data. First, we will try to calculate the visibility of large-scale DEM data with higher accuracy, to fully test the performance, accuracy, and scalability of the algorithm. Second, some other XDraw data decomposition methods have been proposed in the existing research work. We will further organize relevant experiments compared with this algorithm and draw relevant conclusions. Then, we will try to accelerate the calculation of ML-XDraw by heterogeneous computing such as GPU. GPU can speed up computationally intensive algorithms such as XDraw, but Spark's scheduling policies and fault tolerance mechanism do not fully support multi-core computing hardware such as GPU. We will further explore acceleration solutions that integrate the Spark distributed framework with multi-core computing hardware such as GPUs. Finally, we will refine the comparison with existing computing frameworks. Some current computing frameworks support viewshed analysis for large-scale grid data, and we will further supplement their comparative experiments with the algorithm in this article and support relevant conclusions in this article.

**Author Contributions:** Conceptualization, J.D. and J.Z.; methodology, J.D.; software, J.Z.; validation, J.D. and J.Z.; formal analysis, J.D.; investigation, J.D.; resources, J.Z.; data curation, J.D.; writing—original draft preparation, J.D.; writing—review and editing, J.Z.; visualization, J.D.; supervision, J.D.; project administration, J.D.; funding acquisition, J.Z. All authors have read and agreed to the published version of the manuscript.

**Funding:** This research was funded by National Natural Science Foundation of China under grant No.41871304.

**Data Availability Statement:** Data available in a publicly accessible repository. The data presented in this study are openly available in https://data.gov.au/data/dataset/9a9284b6-eb45-4a13-97d0-91bf25f1187b, accessed on 1 November 2022.

**Acknowledgments:** The authors would like to thank C.Chen and S.Zhao from the China University of Geosciences for their earlier research work.

**Conflicts of Interest:** The authors declare no conflict of interest.

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
