# Peer review of "A Multi-Level Distributed Computing Approach to XDraw Viewshed Analysis Using Apache Spark"

_remotesensing, doi:10.3390/rs15030761_

Round 1

Reviewer 1 Report

This paper presents an interesting way to improve the XDraw algorithm for generating viewsheds. This research is very relevant to many fields of knowledge, especially those related to landscape analysis. I would like to comment on some of the content presented.

1. In addition to the algorithms mentioned in the introduction for generating visibility graphs, there is also an algorithm based on Whitted's ray tracing. Some of its elements and applications were presented in 2017 at the Digital Landscape Architecture conference. As far as I follow the literature it has not been tested in such large-scale applications.

2. The presentation of the algorithm itself is well described, however sometimes difficult to understand for those less familiar with technical language.

3. the large numbers in the tables are very difficult to comprehend (especially in table 4). Table 3 is confusing due to the combination of times and dimensionless units. The use of the decimal system to express time is also problematic for the readability of the results.

4. The major shortcoming in my opinion is the lack of extended studies of the accuracy of the results generated by the proposed algorithm. They basically consist of Figure 10 and Table 4 with the aforementioned figure raising quite a few questions. In it one can see a comparison of the viewsheds created by the Serial XDraw and ML-XDraw algorithms. At first glance, they look very similar, but after closer inspection, rather significant differences are apparent. For the first pair it is mainly the area in the north, for the second - in the west, for the third - a large area in the central part. Is the CR ratio really around 97%? The viewsheds shown suggest that the ML-XDraw algorithm tends to mark more areas as visible than the Serial version. I would suggest comparing the images with each other. As the viewsheds are binary images it is easy to calculate the absolute difference between them, which will indicate where the differences are in the visibility computation. It may also help to calculate the mean brightness of the individual viewsheds, which will clearly indicate the differences quantitatively. In my opinion, the difference seen for the Flat point is very significant and may disqualify the presented solution in some applications. It would also be very helpful to mark the observation points on the presented viewsheds.

In conclusion, while the algorithm itself and the way it is presented is of a high standard, more in-depth studies should be carried out regarding the correctness of the generated results.

Best regards to the Authors.

Reviewer 2 Report

An important area of research is covered in the manuscript. The manuscript is academically sound. The manuscript's phrasing has to be changed, though. It is also important to check the references included in the work. If the authors had included a comparison of similar works by other writers in terms of statistics and accuracy, this manuscript would have been better. More comparisons with other methods are needed, and it should be explained what makes ML-XDraw special.

Reviewer 3 Report

DEM-based watershed calculation is a classic and basic topic in digital terrain analysis. However, related algorithm is always of high time-cost and thus distributed/parallel computing is always involved. In this sense, this manuscript tried to address a hot issue in the community. The overall novelty is satisfied as I believe, as well as the scientific meanings. Nevertheless, I listed some recommendations and comments as follow for an improvement.

1. The second paragraph of introduction section introduced a very hot issue in viewshed calculation, data volume vs. time efficiency. This is true and mainly due to the huge improvement on the spatial resolution of DEM data in recent year. Laser-scanning-based data generation could give the users a very fine resolution data. However, from the following text, I see the authors in fact used a traditional resolution data, but only with big coverage area (in other words, huge data volume). So, the problem introducing is not well-established in this sense.

2. Similar to the above-mentioned, the next two paragraphs introduced the main categories of DEM-based algorithms, including R3, R2, and XDraw. It would be much better to give a very brief introduction about the basic idea/principle of each kind of algorithm, as well as the corresponding references. This will greatly help the readers to get a clear picture about how and why these three methods are different, especially for those who are not familiar with detailed algorithm designs. Some illustration figures will be also helpful.

3. I am very happy to see a clear description about how proposed enhanced XDraw was designed in section 2.

4. In section 3, it would be better to give some detailed description about the test datasets, for example, extent (with enlarged window of colored height for details) and height histogram (because terrain morphology may influence the algorithm).

5. It would be much better if the authors can provide an accuracy table and some figures of extraction result with an overview and enlarged details.

6. For accuracy assessment, DEM4’s result is missing. Same problem also occurred in whole result section. Why only DEM3’s result was given in Fig. 10.

Round 2

Reviewer 1 Report

I would like to thank the Authors for the detailed explanations and improvements made to the paper. I have one additional question. Are the differences presented in Figure 10 absolute differences? In other words, did the ML-XDraw algorithm mark any cells as visible that were not visible in the Serial algorithm?
The large numbers in Table 4 still need to be clarified to follow. I suggest using commas for digit grouping of thousands.

Reviewer 3 Report

I would like to thank the authors for their revision works and I think they successfully sloved my previous concerns. No more concerns could be raised form my side and I feel the current version is ready to publish.